# Effects of Different Exercises Combined with Different Dietary Interventions on Body Composition: A Systematic Review and Network Meta-Analysis

**DOI:** 10.3390/nu16173007

**Published:** 2024-09-05

**Authors:** Yongchao Xie, Yu Gu, Zhen Li, Bingchen He, Lei Zhang

**Affiliations:** 1Centre for Sport Nutrition and Health, Centre for Nutritional Ecology, School of Physical Education (Main Campus), Zhengzhou University, Zhengzhou 450001, China; 15637175236@163.com (Y.X.); lizhensolazy@163.com (Z.L.); 2Henan Sports Medicine and Rehabilitation Center, Henan Sport University, Zhengzhou 450044, China; 202055433@jbnu.ac.kr; 3Department of Physical Education, South China University of Technology, Guangzhou 510641, China; 202320141950@mail.scut.edu.cn

**Keywords:** aerobic exercise, resistance exercise, caloric restriction, intermittent fasting, ketogenic diet, weight, BMI, body fat percentage, lean body mass

## Abstract

Background: Exercise and dietary interventions are essential for maintaining weight and reducing fat accumulation. With the growing popularity of various dietary strategies, evidence suggests that combining exercise with dietary interventions offers greater benefits than either approach alone. Consequently, this combined strategy has become a preferred method for many individuals aiming to maintain health. Calorie restriction, 5/2 intermittent fasting, time-restricted feeding, and the ketogenic diet are among the most popular dietary interventions today. Aerobic exercise, resistance training, and mixed exercise are the most widely practiced forms of physical activity. Exploring the best combinations of these approaches to determine which yields the most effective results is both meaningful and valuable. Despite this trend, a comparative analysis of the effects of different exercise and diet combinations is lacking. This study uses network meta-analysis to evaluate the impact of various combined interventions on body composition and to compare their efficacy. Methods: We systematically reviewed literature from database inception through May 2024, searching PubMed, Web of Science, Embase, and the Cochrane Library. The study was registered in PROSPERO under the title: “Effects of Exercise Combined with Different Dietary Interventions on Body Composition: A Systematic Review and Network Meta-Analysis” (identifier: CRD42024542184). Studies were meticulously selected based on specific inclusion and exclusion criteria (The included studies must be randomized controlled trials involving healthy adults aged 18 to 65 years. Articles were rigorously screened according to the specified inclusion and exclusion criteria.), and their risk of bias was assessed using the Cochrane risk of bias tool. Data were aggregated and analyzed using network meta-analysis, with intervention efficacy ranked by Surface Under the Cumulative Ranking (SUCRA) curves. Results: The network meta-analysis included 78 randomized controlled trials with 5219 participants, comparing the effects of four combined interventions: exercise with calorie restriction (CR+EX), exercise with time-restricted eating (TRF+EX), exercise with 5/2 intermittent fasting (5/2F+EX), and exercise with a ketogenic diet (KD+EX) on body composition. Intervention efficacy ranking was as follows: (1) Weight Reduction: CR+EX > KD+EX > TRF+EX > 5/2F+EX (Relative to CR+EX, the effect sizes of 5/2F+EX, TRF+EX and KD+EX are 2.94 (−3.64, 9.52); 2.37 (−0.40, 5.15); 1.80 (−1.75, 5.34)). (2) BMI: CR+EX > KD+EX > 5/2F+EX > TRF+EX (Relative to CR+EX, the effect sizes of 5/2F+EX, TRF+EX and KD+EX are 1.95 (−0.49, 4.39); 2.20 (1.08, 3.32); 1.23 (−0.26, 2.71)). (3) Body Fat Percentage: CR+EX > 5/2F+EX > TRF+EX > KD+EX (Relative to CR+EX, the effect sizes of 5/2F+EX, TRF+EX and KD+EX are 2.66 (−1.56, 6.89); 2.84 (0.56, 5.13); 3.14 (0.52, 5.75).). (4) Lean Body Mass in Male: CR+EX > TRF+EX > KD+EX (Relative to CR+EX, the effect sizes of TRF+EX and KD+EX are −1.60 (−6.98, 3.78); −2.76 (−7.93, 2.40)). (5) Lean Body Mass in Female: TRF+EX > CR+EX > 5/2F+EX > KD+EX (Relative to TRF+EX, the effect sizes of CR+EX, 5/2F+EX and KD+EX are −0.52 (−2.58, 1.55); −1.83 (−4.71, 1.04); −2.46 (−5.69,0.76).). Conclusion: Calorie restriction combined with exercise emerged as the most effective strategy for reducing weight and fat percentage while maintaining lean body mass. For women, combining exercise with time-restricted eating proved optimal for preserving muscle mass. While combining exercise with a ketogenic diet effectively reduces weight, it is comparatively less effective at decreasing fat percentage and preserving lean body mass. Hence, the ketogenic diet combined with exercise is considered suboptimal.

## 1. Introduction

With societal advancements and improvements in living standards, certain lifestyle habits that predispose individuals to various chronic diseases have become prevalent. Prolonged sedentary behavior, habitual sleep deprivation, and excessive eating contribute to metabolic disorders, which can lead to obesity or other chronic conditions. Regular physical activity is a well-established method to mitigate these health issues. The myriad benefits of exercise, including muscle mass maintenance, fat reduction, and enhanced metabolic function, are widely recognized. Historically, exercise has been the predominant strategy for weight management and health maintenance. However, in recent years, dietary interventions have gained prominence, offering alternative or complementary approaches to weight management. Notably, the most common dietary strategies include calorie restriction, 5/2 intermittent fasting, time-restricted eating, and the ketogenic diet. Extensive research supports the efficacy of these methods in both animal and human studies, demonstrating significant weight loss and metabolic advantages [1,2,3]. Recent studies have demonstrated that three dietary intervention strategies effectively lower both diastolic and systolic blood pressure [4,5,6,7,8], enhance mitochondrial function [9,10,11], reduce oxidative stress [4,5,12,13], improve insulin sensitivity [14,15,16,17], and foster beneficial modifications in the gut microbiome [18,19,20,21]. Specifically, calorie restriction, with a reduction in energy intake of approximately 15–30%, has been substantiated to decrease body weight, diminish body fat content, prolong lifespan, and lessen the incidence of age-related diseases, including cancer [22,23,24]. Furthermore, moderate calorie restriction (11.9 ± 0.7%) has been shown to significantly enhance quality of life, metabolic health, liver function, skeletal muscle mass, and immune capacity within a two-year period [25,26,27,28,29]. Unlike calorie restriction, intermittent fasting offers flexibility in food selection, integrating seamlessly with existing dietary regimes and presenting a viable, low-maintenance approach to health improvement [30]. Short-term results indicate that 5/2 intermittent fasting and time-restricted eating can achieve a weight reduction of 3–8% [31]. Intermittent fasting shows more pronounced effects in obese and overweight individuals, with an average weekly weight loss of approximately 0.2–0.5 kg [32], whereas individuals of normal weight may experience a reduction of about 0.2 kg per week [33]. Notably, about 75% of weight loss attributable to calorie restriction or intermittent fasting comprises fat mass, with the remaining 25% being lean body mass [34,35,36,37]. The ketogenic diet induces a metabolic state known as ketosis, where ketone bodies, rather than glucose, serve as the primary energy substrate, lowering blood glucose levels and facilitating the use of ketones for energy [38,39]. Additionally, the ketogenic diet has been associated with enhanced immune function [40,41], improved neurological health, and potential efficacy in treating epilepsy and inhibiting tumor growth [42]. Despite these benefits, some dietary interventions may lead to muscle mass reduction, consequently decreasing the basal metabolic rate [7,32], a factor that could be intertwined with overall weight loss [43,44,45]. The reduction in skeletal muscle can lead to various issues, including a lowered basal metabolic rate, impaired enhancement of exercise capacity, and reduced caloric burn [46,47,48]. Research has demonstrated that both aerobic and resistance training are highly effective at preserving lean body mass during weight loss efforts [49]. Subsequently, an increasing number of individuals have explored the combination of exercise and dietary interventions to mitigate the drawbacks of single interventions and achieve synergistic benefits. Recent trials have demonstrated the feasibility of combining exercise with dietary interventions; participants have been able to perform moderate to high-intensity endurance or resistance training during fasting periods of 12 to 36 h without adverse effects after adapting to the fasting regimen [44,45,50,51,52]. Research has shown that exercise surpasses calorie restriction in promoting fatty acid oxidation, elevating high-density lipoprotein levels [3,31,52,53], and enhancing mitochondrial function, while calorie restriction is more effective in reducing body weight [54]. Specifically, calorie restriction reduces oxidative stress, whereas exercise induces oxidative stress in skeletal muscles. These findings suggest that the benefits of exercise and calorie restriction are distinct, indicating that their combination might circumvent the limitations of single interventions and provide greater overall benefits [55,56]. Additionally, studies have indicated that time-restricted eating alone can lead to a reduction in lean body mass. However, research involving individuals who regularly engage in exercise training has found that combining time-restricted eating with resistance training maintains or increases muscle mass while significantly reducing subcutaneous fat mass [44,45]. Several meta-analyses have corroborated that combining exercise and dietary interventions is more effective than either approach alone in improving body composition, glycemic markers, and lipid profiles [57,58,59,60,61,62].

Therefore, an increasing number of studies have focused on combining exercise with dietary interventions to explore whether this approach can yield more effective weight loss outcomes. However, the effects of combining exercise with various dietary interventions differ across studies, and there is currently no comprehensive literature comparing the effects of exercise combined with different dietary interventions. Consequently, this study aims to summarize the outcomes of combining exercise with different dietary interventions and perform a network meta-analysis to investigate the differences in their effects. The objective is to provide evidence-based recommendations for individuals seeking to combine exercise with dietary interventions.

## 2. Methods

### 2.1. Protocol and Registration

This systematic review was conducted in accordance with the Preferred Reporting Items for Systematic Reviews and Meta-Analyses (PRISMA) guidelines [63]. It has been registered in PROSPERO (identifier CRD42024542184).

### 2.2. Search Strategy

A comprehensive search was conducted from inception to May 2024 in the following databases: PubMed, Web of Science, Embase, and Cochrane Library. Only the literature published in English was included. Relevant articles were independently reviewed and evaluated by the author team (Y.X. and L.Z.) based on the inclusion criteria. Disagreements regarding article inclusion or exclusion were resolved through discussion. The search terms are detailed in Appendix A.

### 2.3. Inclusion Criteria and Exclusion Criteria

Inclusion Criteria:(1)Must be a randomized controlled trial and include a group receiving a combination of exercise and dietary intervention.(2)Must include a control group, which could be no intervention, exercise only, or dietary intervention only.(3)Participants must be adults aged 18–65 years.(4)Participants must be healthy individuals.(5)Outcome measures must include at least one of the following: body weight, BMI, fat mass, or fat percentage.

Exclusion Criteria:(1)Studies that are not randomized controlled trials.(2)Non-human studies.(3)Non-original studies, including reviews, letters, case reports, or papers that do not provide accurate and clear data.(4)Studies where participants are children, elderly, or individuals with any diseases.

### 2.4. Data Extraction

Data were extracted from the literature using a pre-designed Excel 2019 spreadsheet, which included the study title, characteristics of study subjects, age, intervention methods, duration of intervention, and quality assessment of the literature. Two authors (Y.G. and Y.X.) independently extracted the data, and any disagreements were resolved through discussion with a third party (L.Z.). For studies where mean values were presented only in graphical format, Web Plot Digitizer (https://automeris.io/, accessed on 1 June 2024) was used to extract the data. If the data were not presented as "Mean ± SD," a standardized template for evidence-based medicine data extraction was used to convert them to the "Mean ± SD" format.

### 2.5. Risk of Bias Assessment

The included randomized controlled trials (RCTs) were assessed for quality using the Cochrane risk of bias tool. The assessment covered seven domains: random sequence generation/allocation concealment (selection bias), blinding of participants/personnel (performance bias), blinding of outcome assessment (detection bias), incomplete outcome data (attrition bias), selective reporting (reporting bias), and other biases. Each section was rated on three levels: low risk, high risk, and unclear. The quality of trials was assessed by two authors (Y.X. and L.Z.), and discrepancies were resolved through discussion with a third reviewer (Y.G.) to reach a consensus.

### 2.6. Statistical Analysis

Review Manager 5.3 was utilized to analyze the literature and generate the Risk of Bias summary and Risk of Bias graph. Stata MP 17 software was employed to conduct the network meta-analysis, assess the consistency between direct and indirect comparisons, compute the League Table for the four interventions, and calculate SUCRA rankings. Additionally, Network Plot, SUCRA Plot, and funnel plots were created. Results were summarized using standardized mean differences (SMD) and 95% confidence intervals (95% CI).

## 3. Results

### 3.1. Search Results and Study Selection

By May 2024, a total of 1752 articles were retrieved from the four databases: 253 from PubMed, 928 from Web of Science, 315 from Embase, and 256 from the Cochrane Library. After removing 367 duplicate articles, 1253 articles were excluded based on their titles or abstracts, leaving 132 articles for full-text review. Of these, eight articles could not be located. Subsequently, 46 articles were excluded for the following reasons: abstract only (*n* = 3), subjects aged over 65 years (*n* = 14), subjects under 18 years old (*n* = 3), participants with high blood pressure (*n* = 10), and studies that did not include exercise combined with dietary intervention (*n* = 16). Ultimately, 78 articles were included in the analysis (Figure 1).

### 3.2. Study Characteristics

A total of 78 articles were included in the meta-analysis (Appendix A). These studies, conducted between 1995 and 2024, included 5219 participants aged 18–65 years. The duration of exercise interventions ranged from 4 weeks to 1 year, encompassing both aerobic exercise and resistance training. Dietary interventions included calorie restriction, 5:2 intermittent fasting, time-restricted eating, and the ketogenic diet. The specific characteristics of the studies are detailed in Appendix A. Among the results, 53 studies reported on exercise combined with calorie restriction, 3 studies reported on exercise combined with 5:2 intermittent fasting, 9 studies reported on exercise combined with time-restricted eating, and 9 studies reported on exercise combined with the ketogenic diet.

### 3.3. Risk of Bias in Included Studies

Of the 78 included studies, 7 could not be assessed for risk level in random sequence generation, 57 could not be assessed for risk level in allocation concealment, and 8 could not be assessed for risk level in incomplete outcome data. All evaluated studies were considered to have an unclear risk level for blinding of participants and personnel, as it is unlikely that the dietary and exercise interventions in the experimental groups could be blinded (Appendix A).

### 3.4. Effects of the Interventions

We first conducted an inconsistency analysis. If significant inconsistency was detected, Bayesian models were employed for further analysis. If integration remained unachievable, subgroup analyses were performed based on age, gender, and type of exercise. When consistency was satisfactory, a Network Plot was created to explore the distribution of main characteristics across studies. Subsequently, a League Table was generated to obtain pairwise comparison data. Finally, a SUCRA Plot was drawn to determine the ranking of different intervention methods.

#### 3.4.1. The Effect of Different Dietary Interventions Combined with Exercise on Body Weight

The Network Plot, League Table, and SUCRA Plot for the effects of different dietary interventions combined with exercise on body weight are shown in Figure 2.

As shown in Figure 2A, most of the included studies primarily compare the effects of calorie restriction combined with exercise versus calorie restriction alone. The groups with calorie restriction combined with exercise, calorie restriction alone, exercise alone, and control groups dominate in terms of participant numbers. Figure 2B summarizes the estimated effect size differences (SMD with 95% CI) for the pairwise comparisons of the 10 intervention methods. Compared to the exercise-alone (EX) group, the calorie restriction combined with exercise (CR + EX) group shows a significant weight reduction, while the weight reductions in the 5/2 intermittent fasting combined with exercise (5/2F + EX), time-restricted feeding combined with exercise (TRF + EX), and ketogenic diet combined with exercise (KD + EX) groups are not significant. Figure 2C presents the SUCRA rankings for the four intervention methods, where the size of the area under the curve represents the effectiveness of the intervention. The larger the area under the curve, the higher the ranking of the intervention’s effectiveness. As shown in the figure, for weight reduction, the ranking of the four interventions is as follows: CR + EX > KD + EX > TRF + EX > 5/2F + EX.

#### 3.4.2. The Effect of Different Dietary Interventions Combined with Exercise on BMI

The Network Plot, League Table, and SUCRA Plot for the effects of different dietary interventions combined with exercise on BMI are shown in Figure 3.

As shown in Figure 3A, most of the included studies primarily compare the effects of calorie restriction combined with exercise versus calorie restriction alone. The groups with calorie restriction combined with exercise, calorie restriction alone, exercise alone, and control groups dominate in terms of participant numbers. Notably, no studies included in the analysis examine the effect of the ketogenic diet (KD) on BMI. Figure 3B summarizes the estimated effect size differences (SMD with 95% CI) for the pairwise comparisons. Compared to the exercise-alone (EX) group, the calorie restriction combined with exercise (CR + EX) group shows a significant reduction in BMI, while the 5/2 intermittent fasting combined with exercise (5/2F + EX) and ketogenic diet combined with exercise (KD + EX) groups do not show significant reductions. The time-restricted feeding combined with exercise (TRF + EX) group even shows a trend towards increasing BMI. Figure 3C presents the SUCRA rankings for BMI reduction, where the size of the area under the curve represents the effectiveness of each intervention. For BMI reduction, the ranking of the four interventions is as follows: CR + EX > KD + EX > 5/2F + EX > TRF + EX.

#### 3.4.3. The Effect of Different Dietary Interventions Combined with Exercise on Body Fat Percentage

The Network Plot, League Table, and SUCRA Plot for the effects of different dietary interventions combined with exercise on body fat percentage are shown in Figure 4.

As shown in Figure 4A, most of the included studies primarily compare the effects of calorie restriction combined with exercise versus calorie restriction alone. The groups with calorie restriction combined with exercise, calorie restriction alone, exercise alone, and control groups dominate in terms of participant numbers. Figure 4B summarizes the estimated effect size differences (SMD with 95% CI) for the pairwise comparisons. Compared to the exercise-alone (EX) group, the calorie restriction combined with exercise (CR + EX) group shows a significant reduction in body fat percentage. The 5/2 intermittent fasting combined with exercise (5/2F + EX) group does not show a significant reduction, while the time-restricted feeding combined with exercise (TRF + EX) and ketogenic diet combined with exercise (KD + EX) groups show a trend towards increasing body fat percentage. Figure 4C presents the SUCRA rankings for body fat percentage reduction, where the size of the area under the curve represents the effectiveness of each intervention. For reducing body fat percentage, the ranking of the four interventions is as follows: CR + EX > 5/2F + EX > TRF + EX > KD + EX.

#### 3.4.4. The Effect of Different Dietary Interventions Combined with Exercise on Lean Body Mass

An inconsistency analysis of lean body mass data revealed significant inconsistency (*p* = 0.004). Factors contributing to this inconsistency include differences in study design, baseline characteristics, measurement methods, publication bias, statistical model selection, and heterogeneity among the included studies. Initially, we attempted to address this by changing the statistical model and using GeMTC software 0.14.3 based on Bayesian models for verification. However, consistency checks could not be performed, indicating a high degree of inconsistency in the data. Suspecting heterogeneity among the included studies, we conducted subgroup analyses. When performing subgroup analysis based on whether the participants were obese, the non-obese group did not form a network due to insufficient data, while significant inconsistency remained in the obese group (*p* = 0.0022). We then conducted subgroup analysis based on intervention duration, whether it was less than or equal to 8 weeks. The results showed good consistency in the subgroup with an intervention duration of 8 weeks or less (*p* = 0.3884), but significant inconsistency in the subgroup with an intervention duration of more than 8 weeks (*p* = 0.0001). Next, we performed subgroup analysis based on exercise type. Significant inconsistency was found in the aerobic exercise subgroup (*p* = 0.001), but good consistency was observed in the resistance exercise and mixed exercise subgroups (resistance exercise: *p* = 0.2355; mixed exercise: *p* = 0.9881). Finally, subgroup analysis based on gender showed good consistency in both subgroups (women: *p* = 0.1759; men: *p* = 0.2033). We concluded that gender might be one of the most important factors affecting lean body mass. Therefore, we conducted network meta-analyses separately for different genders, as shown in Figure 5 and Figure 6.

As shown in Figure 5A, most of the included studies primarily compare the effects of exercise alone (EX) versus time-restricted feeding combined with exercise (TRF + EX) or ketogenic diet combined with exercise (KD + EX). The groups with calorie restriction combined with exercise (CR + EX), EX, and TRF + EX dominate in terms of participant numbers. Figure 5B shows that there are no significant differences between the groups. Compared to the EX group, the CR + EX group shows an upward trend in lean body mass, while the other groups show a downward trend. As shown in Figure 5C, for maintaining lean body mass in men, the ranking of the interventions is as follows: CR + EX > TRF + EX > KD + EX.

As shown in Figure 6A, most of the included studies primarily compare the effects of calorie restriction combined with exercise (CR + EX) versus calorie restriction alone (CR). The groups with CR, CR + EX, exercise alone (EX), and control (CON) dominate in terms of participant numbers. Figure 6B shows that there are no significant differences between the groups. Compared to the EX group, the CR + EX, 5/2 intermittent fasting combined with exercise (5/2F + EX), time-restricted feeding combined with exercise (TRF + EX), and ketogenic diet combined with exercise (KD + EX) groups show a downward trend in lean body mass. As shown in Figure 6C, for maintaining lean body mass in women, the ranking of the effectiveness of the four interventions is as follows: TRF + EX > CR + EX > 5/2F + EX > KD + EX.

## 4. Discussion

Our network meta-analysis included 78 controlled trials to assess the effects of combining exercise with different dietary interventions on body composition. By summarizing the included studies, we ranked the effects of four dietary interventions—calorie restriction (CR), 5/2 intermittent fasting (5/2F), time-restricted feeding (TRF), and the ketogenic diet (KD)—combined with exercise (aerobic, resistance, and mixed exercise) on body composition. Our results indicate that, compared to the other three interventions, combining exercise with calorie restriction is the most effective for reducing weight, BMI, and body fat percentage. For men, combining exercise with calorie restriction is the best approach to maintain lean body mass, while for women, combining exercise with time-restricted feeding is the most effective method for maintaining lean body mass. For exercise types, we conducted a subgroup analysis, categorizing the studies into three groups: aerobic exercise, resistance training, and mixed exercise. The results of the subgroup analysis showed that the trends in the effects of these three types of exercise combined with different dietary interventions on weight, body fat percentage, and lean body mass were generally consistent. However, there were some minor differences. Specifically, for weight reduction, when resistance training was performed, the combination with intermittent fasting was more effective than with calorie restriction or the ketogenic diet (Appendix A).

Dietary and exercise interventions have a significant impact on overall health improvement. Numerous studies have shown that dietary interventions can improve the body’s inflammatory state [64], lower blood pressure, reduce blood lipids, and even alleviate symptoms such as body pain [65]. It is not only the nutritional content that positively affects the body; the quantity and timing of intake can also produce remarkable effects. Consistent with previous research, our study’s results indicate that any dietary intervention is more effective than exercise alone in reducing weight. Peven’s study suggests that even when exercise exceeds the recommended amount in physical activity guidelines, it is not as effective as dietary interventions for weight loss [54], supporting the findings of this NMA. CR + EX is significantly more effective in weight reduction compared to other interventions. The primary reason for this might be that CR + EX clearly defines daily caloric intake, preventing increased caloric intake due to the additional consumption brought by exercise. Although some studies have shown that intermittent fasting does not lead to compensatory eating behaviors [66,67,68,69] and reduces total intake by approximately 25% compared to normal eating patterns [70], when intermittent fasting and the ketogenic diet are combined with exercise, the enhanced metabolism induced by exercise can cause significant fluctuations in intake during feeding periods. This might lead to higher intake than dietary interventions alone, indirectly explaining why the weight reduction effect of exercise is not prominent, possibly due to increased intake. When total caloric intake is restricted, the additional consumption brought by exercise is reflected in weight loss. Subsequently, we conducted a subgroup analysis based on the type of exercise. The results remained consistent with the overall analysis when aerobic and mixed exercises were performed. Only when resistance exercise was performed did TRF + EX prove to be more effective than CR + EX. This finding requires further direct comparative research for verification. For BMI, our study’s results are largely consistent with those for weight. The BMI formula is weight divided by the square of height, so its trend is generally consistent with the trend in weight.

To explore the effects of combining exercise with dietary interventions in the body, it is important not only to focus on weight loss but also to consider changes in body composition, particularly fat reduction [71,72]. Our study shows that for reducing body fat percentage, CR + EX is the most effective, while the other three interventions have similar effects, with KD + EX being the least effective in reducing body fat percentage. The reason for this may be that CR + EX creates a caloric deficit, and exercise helps maintain muscle mass and basal metabolic rate, resulting in these caloric deficits being reflected in reduced fat mass. Eglseer’s network meta-analysis also demonstrated that calorie restriction combined with exercise is the most effective in reducing fat mass [73]. The least effective intervention, KD + EX, might be due to the high fat intake, leading to fat accumulation in the body. Although some studies indicate that the ketogenic diet can promote glucose and fat metabolism [74] and significantly reduce total cholesterol, low-density lipoprotein, and triglycerides [75,76], other research in the literature suggests that the ketogenic diet does not lead to weight gain but can slightly increase fat mass and fat percentage [77]. In our study, when comparing exercise combined with other dietary interventions, KD + EX might be effective in reducing body fat percentage compared to a control group, but the other three interventions are more effective. Interestingly, our study found that for reducing body fat percentage, the effect of TRF + EX is worse than TRF alone, 5/2F + EX is worse than 5/2F alone, and KD + EX is worse than KD alone. There might be three reasons for this outcome. First, it could be because exercise promotes fat absorption and storage during dietary interventions. Since fat is the most efficient energy source, the body may accumulate more fat once an exercise routine is established compared to the dietary intervention alone. Second, changes in body weight might affect body fat percentage. If the reduction in body weight is greater than the reduction in fat mass, the body fat percentage might increase. Regardless, body fat percentage is an important indicator of health, and an increase generally signifies adverse events. Lastly, upon closely examining the articles involving changes in body fat percentage, we found that while the fat data showed good consistency, the direct comparison results between groups did not show significant differences. Therefore, the final results might have been influenced by indirect comparisons, leading to significant findings. This aspect needs further experimental validation.

Studies have shown that the loss of muscle mass and strength poses serious challenges to physical function and carries significant health risks, including increased disability, frailty, and mortality [78,79]. Therefore, maintaining lean body mass is critically important. The results of the four intervention methods on maintaining lean body mass showed significant inconsistency (*p* = 0.004). To address this inconsistency, we first changed the statistical model and used the Bayesian-model-based GeMTC software for verification, but we were unable to perform consistency checks. We then suspected that heterogeneity among the included studies might be a factor. Some studies indicate that the type of exercise and the gender of the subjects greatly influence lean body mass. Furrer et al. demonstrated that resistance training typically leads to a significant increase in muscle fiber size, while endurance training mainly induces mitochondrial biogenesis and metabolic remodeling, resulting in enhanced oxidative capacity without substantially affecting absolute muscle mass [80]. We then conducted subgroup analysis based on the type of exercise. We found that the aerobic exercise subgroup still showed significant inconsistency (*p* = 0.001), but the resistance training and mixed exercise subgroups showed good consistency (resistance training: *p* = 0.2355; mixed exercise: *p* = 0.9881). Consequently, we ranked the effects of dietary interventions combined with resistance or mixed exercise on lean body mass, concluding that CR + EX and TRF + EX are the most effective in maintaining lean body mass. Genton et al. indicated that physical activity could delay lean body mass loss in men but not in women [81], although some research suggests this might be due to the lower intensity of physical activity in women compared to men [82]. If gender significantly impacts lean body mass maintenance, studies with a large difference in the number of male and female participants could lead to significant inconsistency. Our NMA results showed good consistency in both subgroups (women: *p* = 0.1759; men: *p* = 0.2033). Ultimately, we chose to conduct subgroup analyses by gender to avoid significant inconsistency, and our results confirmed the impact of gender on lean body mass maintenance.

For men, the best way to maintain lean body mass is CR + EX. For women, the best way to maintain lean body mass is TRF + EX. For both men and women, the least effective method for maintaining lean body mass is KD + EX. Research indicates that the ketogenic diet can reduce glycogen storage in muscles [83,84]. In muscles, glycogen is not stored alone but is combined with water in a ratio of 1:3 [85]. The ketogenic diet, often high in saturated fats, may lack essential nutrients, particularly carbohydrates, dietary fiber, and micronutrients such as calcium, magnesium, potassium, and vitamins A, B, and B6 [86,87]. Without sufficient carbohydrates in the diet, glycogen accumulation decreases, ultimately leading to a reduction in absolute lean body mass [88]. There is evidence that in untrained individuals, the ketogenic diet may lead to slightly higher lean body mass loss compared to a normal diet [89,90,91]. A meta-analysis assessing the impact of exercise combined with the ketogenic diet on body composition found that the ketogenic diet group had significantly lower lean body mass compared to the non-ketogenic diet group [92]. Some studies suggest that the ketogenic diet increases AMPK activity, which can inhibit mTOR signaling, a key regulator of muscle mass increase [93]. Other research indicates that long-term ketogenic diets may promote inflammation and increase the risk of cellular aging [94]. Thus, our NMA concludes that exercise combined with the ketogenic diet is the least recommended among the four intervention methods.

### Strengths and Limitations

The abundance of articles on dietary interventions provided a solid research foundation for this study, enhancing the accuracy of the results. However, there is a lack of literature directly comparing the four dietary interventions combined with exercise, which makes the results more susceptible to the influence of indirect comparisons.

## 5. Conclusions

The results of this NMA indicate that combining calorie restriction with exercise yields the best overall effects on weight reduction, body fat percentage reduction, and maintenance of lean body mass, making it the optimal choice for improving body composition. Following this is time-restricted eating combined with exercise, which has a slightly lesser impact on body composition compared to calorie restriction combined with exercise. Notably, time-restricted eating combined with exercise is the most effective method for maintaining lean body mass in women. Although combining exercise with a ketogenic diet is effective for weight reduction, it is less effective in reducing body fat percentage and maintaining lean body mass compared to other interventions, making it a less favorable choice.

## Figures and Tables

**Figure 1 nutrients-16-03007-f001:**
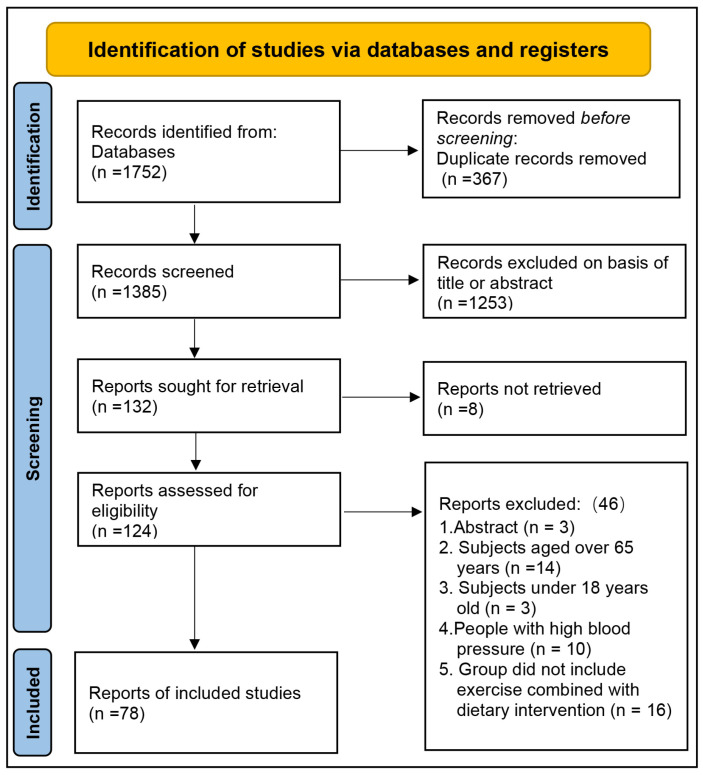
Flow diagram of study selection.

**Figure 2 nutrients-16-03007-f002:**
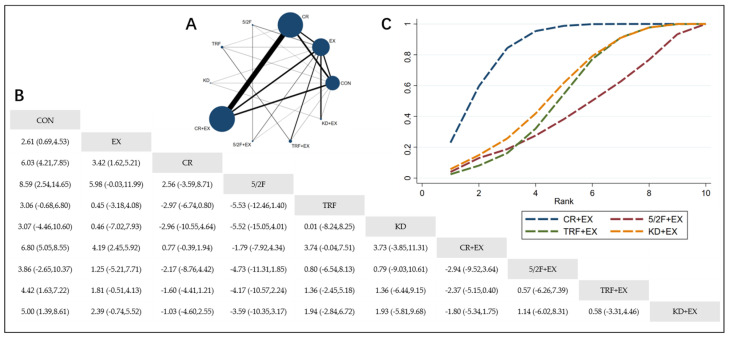
Network Meta-Analysis of Weight: Network Plot, League Table, and SUCRA Plot. (**A**) Network Plot. The size of the nodes is proportional to the sample size of each dietary intervention, and the thickness of the lines corresponds to the number of available studies. (**B**) Pairwise comparison League Table, where the estimated effect size differences (SMD with 95% CI) represent the difference between the intervention on the top and the intervention on the right. (**C**) The SUCRA Plot, where the size of the area under the curve indicates the effectiveness of each intervention.

**Figure 3 nutrients-16-03007-f003:**
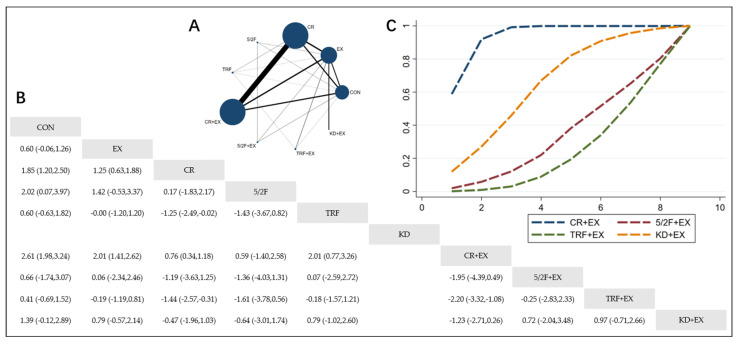
Network Meta-Analysis of BMI: Network Plot, League Table, and SUCRA Plot. (**A**) Network Plot. The size of the nodes is proportional to the sample size of each dietary intervention, and the thickness of the lines corresponds to the number of available studies. (**B**) Pairwise comparison League Table, where the estimated effect size differences (SMD with 95% CI) represent the difference between the intervention on the top and the intervention on the right. (**C**) The SUCRA Plot, where the size of the area under the curve indicates the effectiveness of each intervention.

**Figure 4 nutrients-16-03007-f004:**
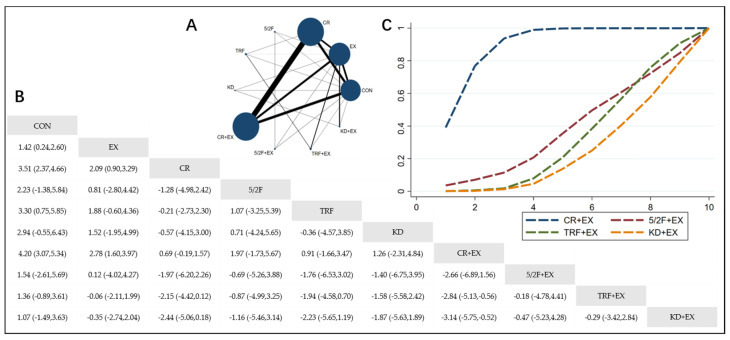
Network Meta-Analysis of Body fat percentage: Network Plot, League Table, and SUCRA Plot. (**A**) Network Plot. The size of the nodes is proportional to the sample size of each dietary intervention, and the thickness of the lines corresponds to the number of available studies. (**B**) Pairwise comparison League Table, where the estimated effect size differences (SMD with 95% CI) represent the difference between the intervention on the top and the intervention on the right. (**C**) The SUCRA Plot, where the size of the area under the curve indicates the effectiveness of each intervention.

**Figure 5 nutrients-16-03007-f005:**
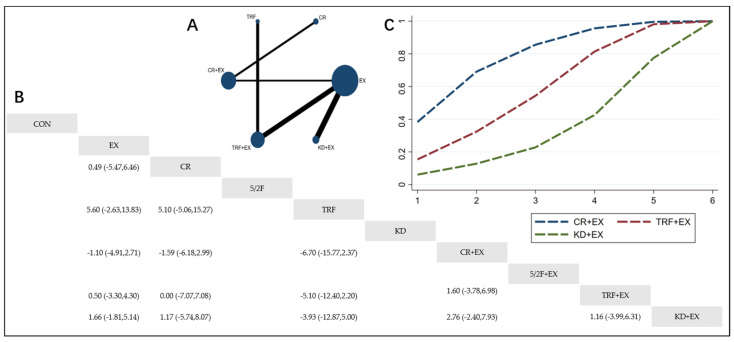
Network Meta-Analysis of male lean body mass: Network Plot, League Table, and SUCRA Plot. (**A**) Network Plot. The size of the nodes is proportional to the sample size of each dietary intervention, and the thickness of the lines corresponds to the number of available studies. (**B**) Pairwise comparison League Table, where the estimated effect size differences (SMD with 95% CI) represent the difference between the intervention on the top and the intervention on the right. (**C**) The SUCRA Plot, where the size of the area under the curve indicates the effectiveness of each intervention.

**Figure 6 nutrients-16-03007-f006:**
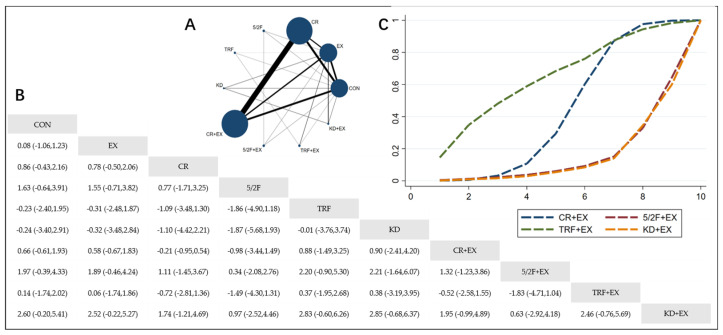
Network Meta-Analysis of female lean body mass: Network Plot, League Table, and SUCRA Plot. (**A**) Network Plot. The size of the nodes is proportional to the sample size of each dietary intervention, and the thickness of the lines corresponds to the number of available studies. (**B**) Pairwise comparison League Table, where the estimated effect size differences (SMD with 95% CI) represent the difference between the intervention on the top and the intervention on the right. (**C**) The SUCRA Plot, where the size of the area under the curve indicates the effectiveness of each intervention.

## Data Availability

Data are available upon request.

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
