# Peer review of "Effects of Different Exercises Combined with Different Dietary Interventions on Body Composition: A Systematic Review and Network Meta-Analysis"

_nutrients, 2024, doi:10.3390/nu16173007_

Round 1

Reviewer 1 Report

Comments and Suggestions for Authors

This paper is in line with the aims of the Journal.

In the abstract, the results of metanalysis should be more described. Moreover, please add resgistration name and number of the Systematic Review and describe inclusion/exclusion criteria.

Keywords: in order to improve the visibility of your article, you should enlarge the number of keywords according to MeSH terms rules.

Introduction is fine and concise.

Methods: the search string and the precise organization of work between researchers must be detailed and clearly reported in this section. About pubmed, why did you search for exercise only and not also for physical activity and synonyms as you did for web of science? Please explain this choice. It is important since, as you know, there are many types of exercise (i.e. physical exercise, therapeutic exercise, etc.) and many types of physical activity, so an unclear use of these terms could have affected your research from the beginning.

Among the files related to this paper, I did not find the PRISMA checklist. Is it possible to verify ot?

Results are clearly presented.

Discussion. This section seems poor. You should enlarge it providing a brief overview about all the possible effect of dietary intervention and exercise in improving general health. To do that, I suggest the following references:

Farì, G., Megna, M., Scacco, S., Ranieri, M., Raele, M. V., Chiaia Noya, E., Macchiarola, D., Bianchi, F. P., Carati, D., Panico, S., Di Campi, E., Gnoni, A., Scacco, V., Inchingolo, A. D., Qorri, E., Scarano, A., & Rapone, B. (2023). Hemp Seed Oil in Association with β-Caryophyllene, Myrcene and Ginger Extract as a Nutraceutical Integration in Knee Osteoarthritis: A Double-Blind Prospective Case-Control Study. Medicina (Kaunas, Lithuania)59(2), 191. https://doi.org/10.3390/medicina59020191

Farì, G., Santagati, D., Pignatelli, G., Scacco, V., Renna, D., Cascarano, G., Vendola, F., Bianchi, F. P., Fiore, P., Ranieri, M., & Megna, M. (2022). Collagen Peptides, in Association with Vitamin C, Sodium Hyaluronate, Manganese and Copper, as Part of the Rehabilitation Project in the Treatment of Chronic Low Back Pain. Endocrine, metabolic & immune disorders drug targets22(1), 108–115. https://doi.org/10.2174/1871530321666210210153619 

Minor corrections:

Line 15: uses instead of employs

Line 121-135: remove points after numbers in the numbered lists

A minor English revision should be useful for this paper.

Best regards and good luck

Comments on the Quality of English Language

Some minor aspects should be better addressed.

I suggest a minor English editing.

Author Response

Comments 1: In the abstract, the results of metanalysis should be more described. Moreover, please add resgistration name and number of the Systematic Review and describe inclusion/exclusion criteria.

Response 1: To enable readers to clearly compare the experimental results, we have added comparative data of the four intervention methods in the abstract (page 1, lines 35-43). Additionally, we have included the registration name and number of the Systematic Review and described the inclusion/exclusion criteria in the abstract (page 1, lines 22-25).

Comments 2: Keywords: in order to improve the visibility of your article, you should enlarge the number of keywords according to MeSH terms rules.

Response 2: We have revised the keywords based on the significance of the terms within the article (page 2, lines 49-50).

Comments 3: Methods: the search string and the precise organization of work between researchers must be detailed and clearly reported in this section. About pubmed, why did you search for exercise only and not also for physical activity and synonyms as you did for web of science? Please explain this choice. It is important since, as you know, there are many types of exercise (i.e. physical exercise, therapeutic exercise, etc.) and many types of physical activity, so an unclear use of these terms could have affected your research from the beginning.

Response 3: When searching for articles in the PUBMED database, we used MeSH terms ("Exercise"[Mesh]), which encompass all exercise-related terms, similar to using multiple exercise-related keywords in the Web of Science database. During our subgroup analysis, we categorized the types of exercise into aerobic, resistance, and combined exercise for further exploration.

Comments 4: Among the files related to this paper, I did not find the PRISMA checklist. Is it possible to verify ot?

Response 4: We sincerely apologize for the oversight. We have now sent the PRISMA checklist as part of the supplementary materials in a compressed file.

Comments 5: Discussion. This section seems poor. You should enlarge it providing a brief overview about all the possible effect of dietary intervention and exercise in improving general health. To do that, I suggest the following references:

Response 5: Thank you for your suggestion. We have expanded the discussion section accordingly (page 10, lines 348-352) (page 17, lines 682-689).

Comments 6: Line 15: uses instead of employs

Response 6: It has been corrected (page 1, line 19).

Comments 7: Line 121-135: remove points after numbers in the numbered lists

Response 7: The corrections have been made (page 3, lines 137-144) (page 4, lines 146-150).

Reviewer 2 Report

Comments and Suggestions for Authors

This study systematically organizes valuable information on how body composition changes when various dietary treatments are combined during exercise.

However, correction and supplementation are required focusing on the following points.

1. The focus of the research must be presented more clearly.

In other words, the types of exercise and diet should be presented more clearly in the topic and purpose of the study.

In this study, the form of exercise is not clearly presented.

Additionally, the diet type also focuses on limiting intake.

2. In addition, the title of the study must be presented specifically in this regard.

3. During the discussion, the types of exercise and diet should also be clearly presented.

4. The conclusion should also be structured more specifically.

Comments on the Quality of English Language

Minor editing of English language required.

Author Response

We feel great thanks for your professional review work on our article. As you are concerned, there are several problems that need to be addressed. The parts we modified are marked in red font in the PDF. According to your nice suggestions, we have made extensive corrections to our previous draft, the detailed corrections are listed below.

Comments 1: The focus of the research must be presented more clearly. In other words, the types of exercise and diet should be presented more clearly in the topic and purpose of the study. In this study, the form of exercise is not clearly presented. Additionally, the diet type also focuses on limiting intake.

Response 1: To help readers better understand the key points of the study, we have added descriptions of the dietary and exercise interventions in the abstract (page 1, lines 14-18). Regarding caloric intake, the included studies categorized restriction into three methods: a fixed daily caloric limit, a percentage of total daily caloric requirements, and a percentage of current total caloric intake. Due to variations among individuals within each study, it was impossible to achieve complete uniformity in summarizing these methods. Therefore, we did not categorize caloric restriction based on intake. For intermittent fasting and ketogenic diets, the majority of studies did not control for caloric intake, and as a result, our manuscript does not specify restrictions on intake.

Comments 2: In addition, the title of the study must be presented specifically in this regard.

Response 2: To make the title more comprehensive, we have revised it (page 1, line 2).

Comments 3: During the discussion, the types of exercise and diet should also be clearly presented.

Response 3: In the discussion, we provided a detailed description of the four dietary approaches and three exercise modalities (page 9, lines 331-335). We also conducted a subgroup analysis based on exercise type and found that the choice of exercise had minimal impact on the optimal dietary approach (page 9, lines 339-347).

Comments 4: The conclusion should also be structured more specifically.

Response 4: Based on your suggestion, we have restructured the conclusion. By organizing it in a sequential manner, we aimed to make the conclusions clearer and easier for readers to understand (page 12, lines 455-464).

Reviewer 3 Report

Comments and Suggestions for Authors

This great study demonstrates the need for caloric restriction in weight loss, as some much-hyped diets claim otherwise. The paper is well structured and comprehensive. I have some remarks:

- Does caloric restriction include the "protein diet"?

- Intermittent fasting exists in various forms, while only one of them has been investigated here.

Author Response

We feel great thanks for your professional review work on our article. As you are concerned, there are several problems that need to be addressed. The parts we modified are marked in red font in the PDF. According to your nice suggestions, we have made extensive corrections to our previous draft, the detailed corrections are listed below.

Comments 1: Does caloric restriction include the "protein diet"?

Response 1: Thank you for providing us with inspiration. This manuscript primarily explores the impact of the timing and quantity of dietary intake, combined with exercise, on body composition. We did not analyze the nutritional composition of the diet in this manuscript. However, based on your suggestion, we plan to investigate the effects of nutritional composition in our future research. We may also explore the interplay between nutritional composition, the timing of dietary intake, and the quantity consumed, and how these factors collectively influence human health.

Comments 2: Intermittent fasting exists in various forms, while only one of them has been investigated here.

Response 2: Before drafting the manuscript, we initially planned to include three of the most widely applied intermittent fasting methods: 5/2 intermittent fasting, alternate-day fasting (1/1), and time-restricted eating (16/8). However, due to the high intensity of alternate-day fasting, we were unable to find suitable literature to include in our study. As a result, we ultimately focused on 5/2 intermittent fasting and time-restricted eating as the primary subjects of our research.

Round 2

Reviewer 1 Report

Comments and Suggestions for Authors

Thank you for the efforts to improve the quality of your paper according to my suggestion.

The article now seems well structured and better organized.

No further corrections are needed.

Reviewer 2 Report

Comments and Suggestions for Authors

You did a good job. However, the consideration for exercise type is insufficient, if possible please comment about his point.